# Molecular Insights of MAP4K4 Signaling in Inflammatory and Malignant Diseases

**DOI:** 10.3390/cancers15082272

**Published:** 2023-04-13

**Authors:** Sunil Kumar Singh, Ruchi Roy, Sandeep Kumar, Piush Srivastava, Saket Jha, Basabi Rana, Ajay Rana

**Affiliations:** 1Department of Surgery, Division of Surgical Oncology, University of Illinois at Chicago, Chicago, IL 60612, USA; 2UICentre for Drug Discovery, University of Illinois at Chicago, Chicago, IL 60612, USA; 3University of Illinois Hospital & Health Sciences System Cancer Center, University of Illinois at Chicago, Chicago, IL 60612, USA; 4Jesse Brown VA Medical Center, Chicago, IL 60612, USA

**Keywords:** mitogen-activated protein kinase, MAP4K4, inflammation, immunity, cancer

## Abstract

**Simple Summary:**

The mitogen-activated protein kinase (MAPK) pathway plays a crucial role in inflammatory and malignant diseases. The MAPK signaling cascade is mediated by various members of the MAP4K, MAP3K, MAP2K, and MAPK families. One of the members of the MAP4K family, MAP4K4, is emerging as a critical player in non-malignant and malignant diseases. MAP4K4 has been investigated for its regulatory role in metabolic and inflammatory diseases and various cancers. Recent understanding of MAP4K4 in malignant diseases suggests a critical role of MAP4K4 in glioblastoma, colon, prostate and pancreatic cancers and associated cachexia. Based on the significant role of MAP4K4 in inflammatory and malignant diseases, it can be utilized as a viable target for therapeutic intervention.

**Abstract:**

Mitogen-activated protein kinase (MAPK) cascades are crucial in extracellular signal transduction to cellular responses. The classical three-tiered MAPK cascades include signaling through MAP kinase kinase kinase (MAP3K) that activates a MAP kinase kinase (MAP2K), which in turn induces MAPK activation and downstream cellular responses. The upstream activators of MAP3K are often small guanosine-5′-triphosphate (GTP)-binding proteins, but in some pathways, MAP3K can be activated by another kinase, which is known as a MAP kinase kinase kinase kinase (MAP4K). MAP4K4 is one of the widely studied MAP4K members, known to play a significant role in inflammatory, cardiovascular, and malignant diseases. The MAP4K4 signal transduction plays an essential role in cell proliferation, transformation, invasiveness, adhesiveness, inflammation, stress responses, and cell migration. Overexpression of MAP4K4 is frequently reported in many cancers, including glioblastoma, colon, prostate, and pancreatic cancers. Besides its mainstay pro-survival role in various malignancies, MAP4K4 has been implicated in cancer-associated cachexia. In the present review, we discuss the functional role of MAP4K4 in malignant/non-malignant diseases and cancer-associated cachexia and its possible use in targeted therapy.

## 1. Introduction

Mitogen-activated protein kinase (MAPK) cascades are key in transducing extracellular signals to cellular responses. The MAPK signaling is a cumulative outcome of multilayered regulation by MAPK upstream regulators, including members of the MAP4K, MAP3K, and MAP2K families (Figure 1). The MAPK signaling is operated via phosphorylation of Serine (Ser) and Threonine (Thr) sites. The MAPKs can phosphorylate their own Ser and Thr residues via autophosphorylation. Additionally, MAPKs can phosphorylate their substrates at Ser and Thr sites, leading to activation or inhibition of downstream signaling [1,2]. In the classical three-tiered MAPK pathways, MAP kinase kinase kinase (MAP3K) activates a MAP kinase kinase (MAP2K), which in turn induces dual Thr and tyrosine (Tyr) phosphorylation and subsequent activation of a MAPK signaling [2,3,4]. The MAPK protein phosphatases inhibit MAPK-mediated signaling via dephosphorylating the phosphorylated MAPK members. The MAPK phosphatases inhibit MAPK phosphorylation at Ser and Thr residues [3,5,6,7]. Three MAPK members are categorized based on differential signature sequences for their activation. (i) Extracellular signal-regulated protein kinases (ERKs) encompass the Thr-Glu-Tyr motif within their activation loop. There are two isoforms of ERK, known as ERK1 and ERK2. (ii) The p38 MAPK has four different isoforms, including p38α, p38β, p38γ, and p38δ MAPKs, which contain the Thr-Ala-Tyr motif in their activation loop. (iii) The stress-activated protein kinases (SAPK), also known as c-Jun N-terminal kinases (JNKs), encompass a Thr-Pro-Tyr motif in their activation loop [8].

There are 22 members of the MAP3K family; including just a few, RAF, MEKK1, Tak1, MLK1, MLK2, MLK3, MLK4, and DLKs are known to regulate downstream MAP2Ks and MAPK-mediated signaling. The MAP3Ks can be regulated by their upstream regulators, MAP4K family members. There are several members of the MAP4K family, including MAP4K1, MAP4K2, MAP4K3, MAP4K4, MAP4K5 (germinal center kinase-related; KHS/GCKR), MAP4K6 (Misshapen-like kinase 1; MINK) and MAP4K7 (NCK interacting kinase; TNIK (MAP4K7) [9,10,11,12,13,14]. The role of the MAP4K family member MAP4K1 has been reported in various diseases, including viral infections and autoimmune and malignant diseases [15,16]. MAP4K2 has been implicated in diabetes-induced retinal vascular dysfunction and cardiotoxicity [17,18]. MAP4K3 has been reported in autoimmune diseases, inflammation, and cancers [19,20,21]. MAP4K4 is the most studied MAP4K family member, known for its role in malignancies [22,23], metabolic and cardiovascular diseases [24], diabetes [25], and antiviral immunity [26]. Another member of the MAP4K family, MAP4K5, has been implicated for its role in inflammatory disease, coronavirus disease 2019 (COVID-19) and malignancies [27,28]. MAP4K6 and MAP4K7 are involved in the Hippo pathway of tissue homeostasis and stress-induced JNK signaling in neuronal cells [29,30,31]. The present review will discuss the functional role of the most-studied MAP4K member, MAP4K4, in inflammatory and malignant diseases, including glioblastoma, colon, prostate, and pancreatic cancers and cancer-associated cachexia, and its possible use in targeted therapy.

## 2. MAP4K4: An Upstream Regulator of MAPKs

MAP4K4 is a member of the germinal cell kinase four (GCK-IV) group [32], which is a Ser/Thr kinase that belongs to the mammalian sterile 20 protein (STE20)/MAP4K family. It controls cellular processes such as cell proliferation and survival, cytoskeletal dynamics, and ion transport [14,33,34,35,36,37]. Initially, the mouse homologue of MAP4K4, NIK, was studied in mice and described as an activator of JNK/SAPK via interacting with MEKK1 and an adaptor protein Nck [9]. Later, the human orthologue of MAP4K4 was identified and cloned [38]. The human MAP4K4 is also known as HPK1/GCK-like kinase (HGK), whereas mouse MAP4K4 is known as Nck-interacting kinase (NIK) [9,38]. In humans, MAP4K4 contains approximately 1280 amino acids encoded with a molecular mass of ~140 KDa, widely expressed in different tissues [9,10,38]. The MAP4K4-encoding gene is located on chromosome 2, position q11.2, and consists of 33 exons responsible for its synthesis [39]. Its orthologues across various species share structural and molecular similarities. Structurally, MAP4K4 comprises several domains, such as an N terminal kinase domain, a coiled-coil domain, a C-terminal hydrophobic leucine-rich citron homology domain (CNH) and two putative caspase cleavage sites [37,39,40,41]. The CNH domain has been shown to be important for protein–protein interactions. For example, in mice, the CNH domain of MAP4K4 interacts with MEKK1 (also known as MAP3K1) and facilitates its interaction with the cytoplasmic domain of β1-integrin receptors [9,42]. Alternative splicing yields five functional isoforms of the MAP4K4 gene with 100% homology. All of these isoforms display kinase and CNH domains but differ in their inter-domain regions. The expression of MAP4K4 is almost ubiquitous in all human tissues [10]; however, it is prominently expressed in the testes and brain [37]. The expression of different isoforms of MAP4K4 is cell and tissue type-dependent [37]. The biological functions of all of the isoforms are unknown; however, it can be speculated that variations in the middle domain could affect MAP4K4 interaction with other proteins, causing different biochemical and physiological functions. The functional importance of MAP4K4 has been established based on genetic evidence from mouse models. The deletion of MAP4K4, either whole-body or endothelial-specific knockout, was lethal. The loss of MAP4K4 resulted in embryonic lethality due to mesodermal and somite development impairment and decreased migration activity of endothelial cells [43,44]. MAP4K4 is overexpressed in many human cancer cell lines and tumors compared to normal tissue [33,45,46]. The overexpression of MAP4K4 has been reported in tumors compared to normal tissues in various cancers [22,23,47]. It has been reported that inactive mutant or dominant negative mutant of MAP4K4 not only suppresses Ras-induced transformation in NIH3T3 cells and rat intestinal epithelial cells but also inhibits the anchorage-independent cell growth and hepatocyte growth factor-stimulated epithelial cell invasion. Furthermore, the knockdown of MAP4K4 by small interfering RNA inhibits the tumor cell migration and invasion of various cancer types and malignant melanoma [46].

## 3. MAP4K4 Signaling in Inflammation

Several studies have reported MAP4K4 as an inflammation-related kinase (Figure 2). MAP4K4 could direct its effect through three possible transduction pathways: JNK, p38 MAPK, and ERK1/2. MAP4K4 is also activated by transforming growth factor β-activated kinase (TAK1), a member of the MAP3K family [10]. The C-terminal regulatory domain of MAP4K4 is believed to regulate MAP4K4 activity. Full activation of JNK by MAP4K4 requires both MAP4K4′s kinase activity and the C-terminal regulatory domain that mediates the association of MAP4K4 with MEKK1 (mitogen-activated protein kinase kinase kinase 1) [9]. Hematopoietic progenitor kinase 1 (HPK1) is a mitogen-activated protein kinase kinase kinase kinase 1 (MAP4K1), which is a hematopoietic-specific mammalian STE20-like protein serine/threonine kinase that activates MAPK signaling via its downstream MAP3K proteins (MEKK1, MLK3, and TAK1), leading to activation of MAPK JNK [48]. The knockdown of MAP4K4 by specific siRNA does not affect the phosphorylation of p38 or ERK but significantly inhibits the JNK phosphorylation [1]. In vitro expression studies have shown that MAP4K4 activates MEKK1, MKK4, and JNK cellular signaling cascade. JNK is activated by various environmental stresses and extracellular stimuli, causing the production of tumor necrosis factor-α (TNF-α), epidermal growth factor (EGF), platelet-derived growth factor, transforming growth factor β (TGF-β) and lysophosphatidic acid [49,50]. ERK1 and ERK2 were the first MAPK subfamilies investigated to understand the MAPK signaling pathway. After that, most of the studies focused on screening the role of stress-activated kinases, especially p38/SAPK MAPK and JNK MAPK. There is a very high fidelity and selective adaptation of each MAPK module required to translate extracellular signals into physiological responses, and there are even sequence similarities. Activation of JNK is required for multiple physiological processes such as apoptotic regulation, tumorigenesis, and/or inflammation. JNK-mediated phosphorylation of c-Jun leads to increased activity of the AP1 transcription factor. Substantial evidence in different mammalian and fly cell systems revealed that the function of MAP4K4 is closely associated with the TNF-α-induced JNK signaling pathway [10,51,52]. Moreover, in human cell lines, activation of JNK by MAP4K4 is reported by TNF signaling [9]. TNF stimulates MAP4K4 mRNA expression through its tumor necrosis factor receptor 1 (TNFR1), leading to the activation of the transcription factors c-Jun and activating transcription factor-2 (ATF2) [53]. Interestingly, this occurs in a loop where MAP4K4 stimulates TNF-α signaling and its production [10], which causes a further increase in MAP4K4 expression via the transcription factors c-Jun and ATF2 [32,53]. TNFα plays an important role in regulating the expression of MAP4K4 and has a more substantial effect on the phosphorylation of c-Jun and ATF2 than other inflammatory cytokines, such as IL-1β, which cannot modify the mRNA expression of MAP4K4 [53]. Still, studies could not determine the intermediate kinases implicated in the activation of JNK by MAP4K4 and the downstream factors through which MAP4K4-JNK mediates its effects [33]. Further, elucidating the MAP4K4–JNK axis should provide mechanistic insight into understanding inflammatory signaling.

## 4. MAP4K4 Signaling in Vascular Inflammation and Atherosclerosis

Clinical complications of atherosclerosis, such as myocardial infarction (MI), stroke, and peripheral arterial disease, represent the leading causes of mortality and morbidity [54]. Atherosclerosis is a physical ailment accompanied by a chronic inflammatory response at susceptible sites in the walls of major arteries. It is facilitated by perturbed vascular flow and oxidized lipoprotein-mediated vascular inflammation [55,56]. Since MAP4K4 is overexpressed in endothelial cells (ECs), the lining of the circulatory and lymphatic vasculatures, this suggests the possible role of MAP4K4 in lymphatic vessel formation and function [24]. In ECs, inflammatory cytokines, including TNF-α, induce the expression of several inflammatory genes, including genes for leukocyte adhesion molecules and chemokines [57]. Therefore, all of the published reports support MAP4K4 as a proinflammatory kinase mediating the deleterious functions of TNF-α [58,59]. MAP4K4 promotes endothelial permeability, enhancing the inflammatory response to lipid-mediated vascular damage leading to atherosclerosis [60]. Deleting MAP4K4, specifically in ECs, by gene silencing and gene ablation experiments in Apoe-/-mice resulted in reduced aortic macrophage accumulation and chemokine content. MAP4K4 silencing attenuated EC adhesion molecule expression and leukocyte recruitment to atherosclerotic plaque areas. Moreover, this study indicated that silencing MAP4K4 can ameliorate inflammatory responses by reducing TNFα-mediated deleterious effects [61]. In addition to maintaining endothelial functionality, MAP4K4 has been reported to play an important role in angiogenesis [24]. It could also contribute to vascular functions that influence insulin sensitivity and glucose homeostasis, including insulin delivery to skeletal muscle and modulation of inflammation in adipose tissue [62,63]. These reports suggest that MAP4K4 could be a potential target for treating vascular inflammation and atherosclerosis; however, further in-depth studies are still required to understand the role of other proinflammatory mediators.

## 5. MAP4K4 Signaling in Adaptive Immunity

MAPK pathways are crucial for regulating T cell function involving proliferation and activation modulated by T cell receptor (TCR) and CD28 costimulatory molecules. MAP4K4 has been reported to regulate complex TCR responses of primary T cells via NF-kB and MAPK pathways. The CD4^+^ T cell can be activated by the engagement of its receptors with epitope(s) present in antigenic proteins and with external stimuli. The naïve CD4^+^ T cells undergo proliferation and differentiation upon activation. The activated CD4^+^ T cells can be differentiated into their subsets, including T helper 1 (Th1), T helper 2 (Th2), and T helper 17 (Th17). These CD4^+^ T cell subsets are known to produce cytokines/chemokines required for T cell-mediated immune reactions. The regulatory T cells (Treg) are an important subset of CD4^+^ T cells, needed for T cell homeostasis and associated with immune tolerance [64,65]. As MAP4K4 is known to regulate the expression of the proinflammatory cytokine TNF-α, it was demonstrated that activation of Jurkat T cells by antigen-presenting cells (APCs) loaded with antigen(s) leads to MAP4K4-mediated TNF-α production [66]. The MAP4K4 deficiency leads to a substantial reduction in T cell activation, proliferation, and production of interleukin 2 (IL-2) and interferon-γ (IFN-γ). However, MAP4K4 inhibition in T cells increased the expression of transcription factor Foxp3 in peripheral Tregs. The role of MAP4K4 has been established in CD4^+^ T cell proliferation. T cells were activated with PMA/Ionomycin in MAP4K4 sufficient and MAP4K4 deleted cells, where the expansion of CD4^+^ T cells was compromised upon MAP4K4 deletion [67]. There is a need for in-depth mechanistic studies on the role of MAP4K4 in immunity using in vivo systems.

## 6. The Regulatory Functions of MAP4K4 in Malignant Diseases

MAP4K4 is overexpressed in many human cancers compared to normal tissues [23]. The first evidence suggesting the role of MAP4K4 in modulating cellular transformation, adhesion, and invasion came from the study by Wright et al., 2003; overexpression of MAP4K4 was reported in 40 of the NCI-60 human tumor cell lines [37]. The highest upregulation in MAP4K4 expression, using a National Cancer Institute (NCI) tumor panel, was reported in glioblastoma cancer cell lines [37]. The overexpression of MAP4K4 is reported in various cancers, including lung, liver, prostate, ovarian, and pancreatic cancers [33,37,46,68]. MAP4K4 has been reported to play a critical role in cell migration, invasiveness, and adhesion in various types of cancers (Figure 3). The migration and motility of cells are associated with invasive and metastatic properties of cancer cells [69]. Several signaling cascades culminate in cell migration and invasion, which are involved in cancer progression and are activated by either overexpression or different types of mutations (Figure 2) [33]. The JNK signaling is one of the crucial mediators associated with MAP4K4 pathways to increase metastasis, contributing to stress responses, cell proliferation, apoptosis, and tumorigenesis [33]. The following sections describe the role of MAP4K4 in colorectal and hepatocellular carcinoma, gastric cancer, and prostate and PDAC cancers.

### 6.1. Role of MAP4K4 in Colorectal Cancer

The cancer of the colon or rectum or both is known as colorectal cancer (CRC). CRC has a significant role in cancer-related death worldwide, and currently, it is ranked as the fourth leading cause of cancer-related mortality [70]. The severity of CRC is mainly because of increased evidence of uncontrollable cancer growth and metastasis [70] and drug resistance. Therefore, finding a novel targeted therapy will help obtain therapeutic solutions. The role of MAP4K4 in CRC has been established as that of a regulator of cancer cell proliferation in vitro and in vivo models. The knockdown of MAP4K4 using MAP4K4-specific siRNAs in CRC cancer cells showed inhibition in cellular proliferation [71]. These preclinical observations indicate that targeting MAP4K4 in CRC will have a robust antitumor effect and could be a potential candidate for clinical trials in the near future.

### 6.2. Role of MAP4K4 in Hepatocellular Carcinoma

Similarly, hepatocellular carcinoma (HCC) is the third leading cause of cancer mortality worldwide, with more than 600,000 new cases diagnosed yearly. Early stages of HCC are potentially curable by resection, liver transplantation, or percutaneous treatment. Due to the aggressive nature of HCC and drug resistance, the five-year patient overall survival is restricted to about 60% [72]. Sorafenib, which can target multiple kinases, is prescribed in the clinic for the advanced stage of HCC. Sorafenib helps improve the condition of HCC patients; however, increasing evidence of sorafenib resistance is a clinal challenge for HCC therapy [73,74]. These complications with sorafenib indicated a need for a molecular target for HCC therapy. Interestingly, MAP4K4 was overexpressed in HCC patients with poor prognoses [45]. A study using human HCC cell lines, including HepG2 and Hep3B, showed that inhibition of MAP4K4 decreases HCC cell growth by inhibiting cellular proliferation and inducing cellular apoptosis. Further study showed that inhibition of MAP4K4 using shRNA inhibits the cell cycle in S-phase to prevent cellular proliferation [45]. These results indicate that MAP4K4 can potentially control HCC proliferation and survival and could be used as targeted therapy in HCC.

### 6.3. Role of MAP4K4 in Gastric Cancer

Gastric cancer (GC) is the second leading cause of cancer-associated mortality worldwide [75]. The analysis of the TCGA dataset for stomach adenocarcinoma suggests that the mRNA expression of MAP4K4 is overexpressed in GC tumors compared to normal tissues, indicating that perhaps MAP4K4 has oncogenic features. A study where GC and matched normal tissues were used to evaluate the expression of MAP4K4 showed that in 72% of GC patients, MAP4K4 is overexpressed compared to its levels in matched normal tissues [76]. A preclinical study to understand the role of MAP4K4 in GC proliferation using BGD-823 revealed that the knockdown of MAP4K4 induces cell cycle arrest in the G1 phase, leading to decreased cellular proliferation [76]. Zhang et al. reported overexpression of MAP4K4 kinase in gastric tumor samples. The overexpression of MAP4K4 was associated with gastric cancer’s invasion, progression, and metastasis, suggesting that MAP4K4 could serve as a prognostic marker for gastric cancer [77]. The regulatory role of MAP4K4 in GC cellular proliferation suggests that further in-depth study of MAP4K4 in GC is required to explore MAP4K4 as a therapeutic target.

### 6.4. Role of MAP4K4 in Lung Adenocarcinoma

Lung cancer is the most prevalent cancer type worldwide and accounts for the highest cancer-related mortality. Broadly, lung cancer is categorized into two categories: small cell and non-small-cell lung cancer. A comparison of MAP4K4 mRNA and protein expressions in lung tumors and normal/non-tumorous lung tissues revealed that MAP4K4 mRNA/protein expression was higher in tumors [78]. The increased expression of MAP4K4 in lung tumors is negatively associated with prognosis [78]. The patients with higher expression of MAP4K4 in tumors showed shorter overall survival and recurrence of disease [78]. These preliminary findings suggest a critical role of MAP4K4 in lung cancer, and further mechanistic understanding of MAP4K4 in lung cancer is required to open the therapeutic approach to lung cancer.

### 6.5. Role of MAP4K4 in Prostate Cancer

The malignant disease of the prostate gland is known as prostate cancer (PCa). Emerging evidence has shown that MAP4K4 plays an important role in PCa progression and survival [79]. An in vitro study using PCa cells showed that MAP4K4 knockdown decreases cellular proliferation [79]. Kim et al. showed that partial knockdown of MAP4K4 levels or inhibition of its kinase activity replaces SV40 Small T antigen (ST) in cell transformation, suggesting that MAP4K4 is a key PP2A substrate for cell transformation [80]. These results suggest a significant role of MAP4K4 in PCa cell proliferation, and MAP4K4 might be a potential candidate for PCa treatment in the future.

### 6.6. Role of MAP4K4 in Pancreatic Cancer

Pancreatic cancer is one of the deadliest malignancies, with a median OS of around six months [81]. Most pancreatic cancers are pancreatic ductal adenocarcinoma (PDAC) with KRAS mutation and lack therapeutic interventions [81]. Targeted therapy for PDAC is urgently needed, and interestingly, preliminary findings showed a potential role of MAP4K4 in PDAC growth, metastasis, and recurrence [46]. Further analysis for MAP4K4 correlation with prognosis showed that overexpression of MAP4K4 is correlated with poor prognosis [46]. These results suggest that MAP4K4 is a prognostic marker in PDAC. The expression of MAP4K4 is negatively regulated by microRNA-141 (miR-141) in pancreatic cancer [82]. Mechanistic studies suggest that either ectopic expression of miR-141 or knockdown of MAP4K4 attenuates pancreatic tumorigenesis [82]. The downregulated MAP4K4 expression promotes the antitumor effect and decreases drug resistance in pancreatic cancer [82]. These results indicate that MAP4K4 might play an oncogenic role in pancreatic cancer (Table 1). MAP4K4 regulates pancreatic cancer cell proliferation via its downstream target, MLK3, and perhaps other targets [23]. MAP4K4 phosphorylates MLK3 at Threonine residue T738. The MAP4K4–MLK3 axis promotes cell proliferation, colony formation, and cell migration in pancreatic cancer [23]. MAP4K4-specific pharmacological inhibitor GNE-495 [83] showed an antitumor response in in vitro and in vivo models of pancreatic cancers [23]. The compound F389-0746 has been identified as a potential MAP4K4 inhibitor that could inhibit its activity with an IC50 value of 120.7 nM. Compared to gemcitabine, F389-0746 showed better anticancer activity, including tumor growth inhibition in a xenograft pancreatic model [84]. Overall, the role of MAP4K4 in pancreatic malignancy is crucial and might be used for future therapeutic targets. However, in-depth mechanistic studies are still required to establish MAP4K4 as a key player in pancreatic cancer. MAP4K4 is known to regulate MLK3 activity [23], and MLK3 has a significant role in tumor progression and T-cell response [85,86,87]; therefore, it will be essential to establish the in-depth role of MAP4K4 and its downstream signaling in malignant and non-malignant diseases.

## 7. Role of MAP4K4 in Cancer Cachexia

Cancer-associated cachexia is a wasting syndrome in cancer patients characterized by loss of skeletal muscles, loss of healthy adipose tissues, and loss of body weight [88]. Cancer cachexia is a syndrome mediated by abnormal metabolic and inflammatory signaling from the host and tumors [89,90]. This abnormal metabolic and inflammatory signaling causes protein degradation in skeletal muscle and fat loss, leading to body weight loss [89,90]. Cancer-associated cachexia is common in advanced-stage cancer, including gastric, pancreatic, lung, and colorectal cancers [91,92].

Inflammatory signaling is one of the leading causes of cancer-associated cachexia. For example, a proinflammatory cytokine, TNF-α, induces chemokines and monocyte chemoattractant protein-1 (MCP-1) production in pre-adipocytes and adipocytes cells. The increased expression of MCP-1 promotes monocyte recruitment, ultimately leading to strong inflammatory reactions [93]. TNF-α produced by macrophages within the white adipose tissue of cachectic subjects probably requires MAP4K4-JNK-AP1 signaling pathways. Macrophage-derived TNF-α enhances lipolysis and downregulates peroxisome proliferator-activated receptor γ (PPARγ)-mediated triglyceride (TG) biosynthesis and storage in adipocytes [94]. The proinflammatory cytokine TNF-a also negatively regulates PPARγ function in cancer cachexia. TNF-a negatively regulates PPARγ expression (both mRNA and protein) via NF-kB activation. Not only expression, but TNF-a also regulates the activity of PPARγ. TNF-a inhibits PPARγ activation in tumor and tumor-infiltrating macrophages. The inactivated PPARγ compromises TG status in adipocytes [94]. Interestingly, TNF-a has also been reported to regulate skeletal muscle differentiation [95,96,97]. Mouse myogenic in vitro models using C2C12 cells indicated a possible role of MP4K4 in TNF-a mediated skeletal muscle degeneration during cachexia [95,96,97]. MAP4K4 is reported to have a regulatory role in the generation of skeletal muscles. MAP4K4 regulates muscle generation via myogenic factor 5 (Myf5). One study suggests that inhibition of MAP4K4 may restore muscle degeneration during traumatic and dystrophic injuries in muscles [98]. The mechanistic study revealed that MAP4K4 activity is crucial for muscle degeneration [98]. Interestingly, the MAP4K4 inhibitor DMX-5804 was able to reverse abdominal aortic aneurysms (AAAs) induced by loss of RhoA in vascular smooth muscle cells [99]. These preclinical findings suggest that MAP4K4 might be a viable target for cancer-associated cachexia.

## 8. MAP4K4 as a Therapeutic Target

The expression patterns of a particular set of genes, also known as the gene signature, are important to explore malignancies’ phenotypes, patients’ overall survival, and potential therapeutic outcomes [100,101,102]. Based on gene signature analysis, targeted therapy could be proposed to conquer cancer effectively. Novel approaches are needed to detect cancer in the very early stages because cancer patients’ survival rate has not improved in recent years. New approaches are required to improve survival and decrease morbidity (Figure 4).

Improved approaches to understanding cancer development are crucial for developing therapeutic targets. Interestingly, animal models are available, including mouse models that recapitulate human cancer. Studies have shown that MAP4K4 broadly regulates various biological functions and is implicated in disorders leading to disease conditions and cancer. Therefore, this could be a great target in cancer immunotherapies and the development of small molecules to inhibit tumor growth and invasiveness. Co-depletion of MAP4K4 and striatin 3 (STRN3) caused a near-complete eradication of medulloblastoma and completely abrogated cell dissemination in vitro [103]. A study determined the pharmacological inhibition of MAP4K4 using PF-06260933, a selective small-molecule inhibitor against MAP4K4 kinase activity on MAPK signaling in response to TNF-α. Its application to human aortic ECs ameliorates atherosclerosis progression and promotes regression in Apoe−/−mouse models [61]. Another selective MAP4K4 potential inhibitor, GNE-220, has been shown to target pathological angiogenesis. Findings by Vitorino et al. on how MAP4K4 regulates endothelial cell membrane retraction showed that GNE-220 reduced phosphorylation of ERM+ (ezrin, radixin, moesin) retraction fibers. It inhibited MAP4K4-dependent talin replacement by C-terminally phosphorylated moesin from the intracellular domain of integrin β1 [44]. However, the major drawback of GNE-220 is its high brain penetration ability, which highlights the need to optimize its molecular properties to limit brain exposure. Therefore, studies are under way to identify a new class of isoquinoline and naphthyridine-based MAP4K4 inhibitor, GNE-495, that not only has reduced brain exposures but also maintains potent activity and better kinase selectivity. The potency of GNE-495 has been demonstrated for its in vivo efficacy in a retinal angiogenesis model in inducible MAP4K4 knockout mice after intraperitoneal administration. Results indicated that GNE-495 delayed retinal vascular outgrowth and induced abnormal retinal vascular morphology. These findings suggest that GNE-495 can potentially recapitulate the retinal vascular phenotypic defects in MAP4K4 inducible knockout mice [83]. The pharmacological inhibitors of MAP4K4 offer immense possibilities for targeting MAP4K4 in various diseases, including cancers in the clinical setting.

## 9. Conclusions

In summary, MAP4K4 signaling represents a potential prospect for treating inflammatory and malignant diseases. As discussed in this review, MAP4K4 signaling is dysregulated in many inflammatory diseases and malignancies and has been associated with tumor progression, metastasis, and poor prognosis. Identifying new drugs that target the MAP4K4 signaling pathway will be crucial for future therapeutic approaches.

## Figures and Tables

**Figure 1 cancers-15-02272-f001:**
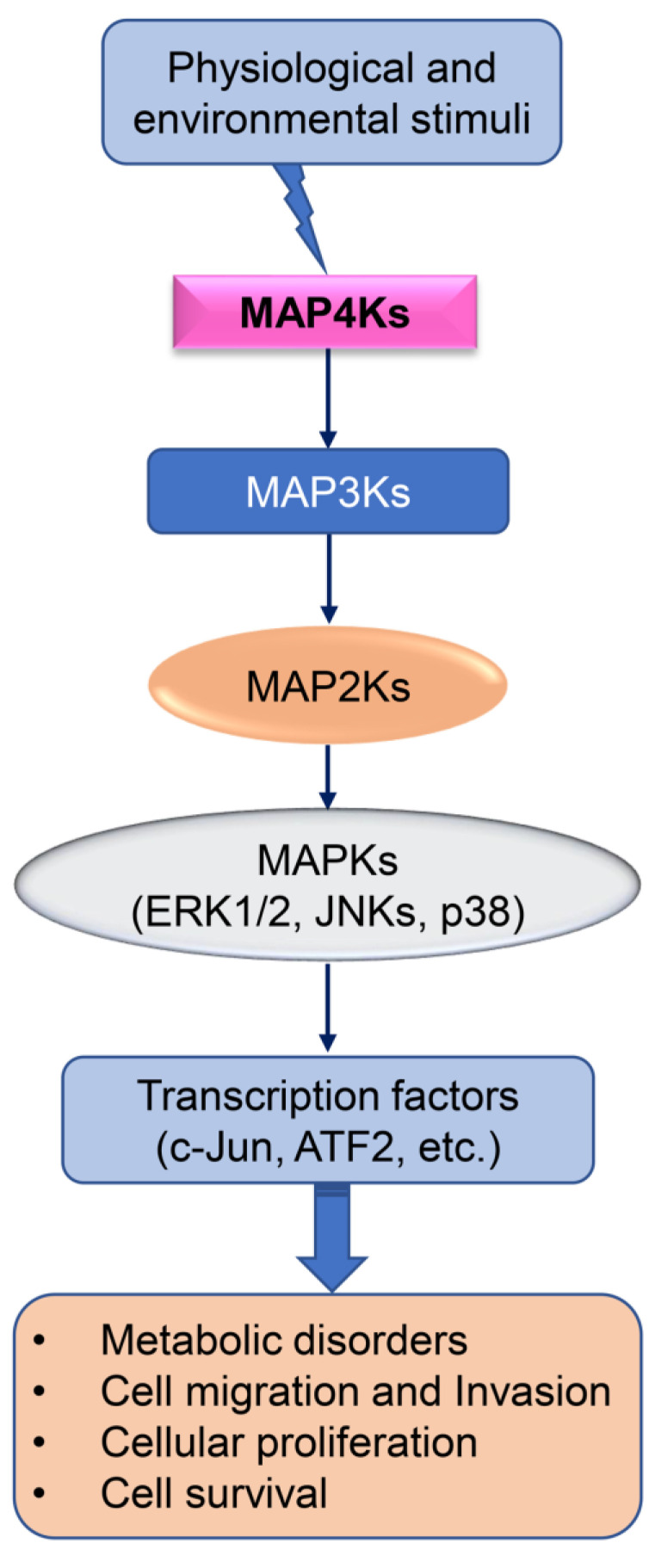
Schematic diagram of MAPK signaling. Physiological and environmental stimuli induce downstream effectors, leading to cellular responses.

**Figure 2 cancers-15-02272-f002:**
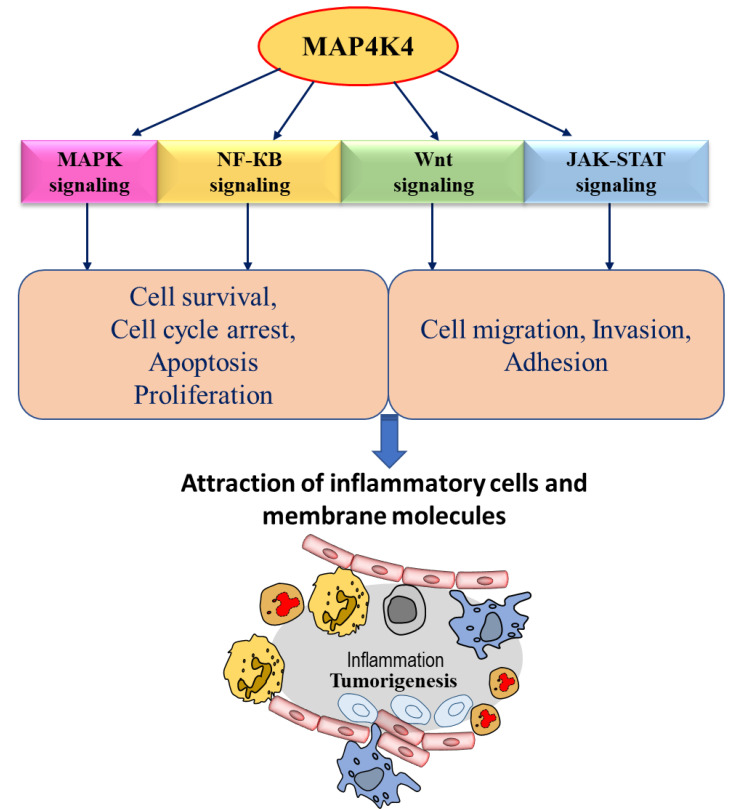
Schematic diagram representing MAP4K4 downstream signaling in cancer. MAP4K4 regulates different biological outcomes through different cell signaling pathways, implicating that MAP4K4 exerts its role in regulating the tumorigenesis process.

**Figure 3 cancers-15-02272-f003:**
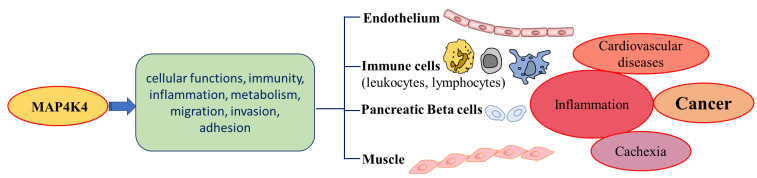
Involvement of MAP4K4 in a variety of regulatory pathways. Effects of MAP4K4 in various metabolic processes involving leukocytes, pancreatic beta cells, and muscles lead to different cancers.

**Figure 4 cancers-15-02272-f004:**
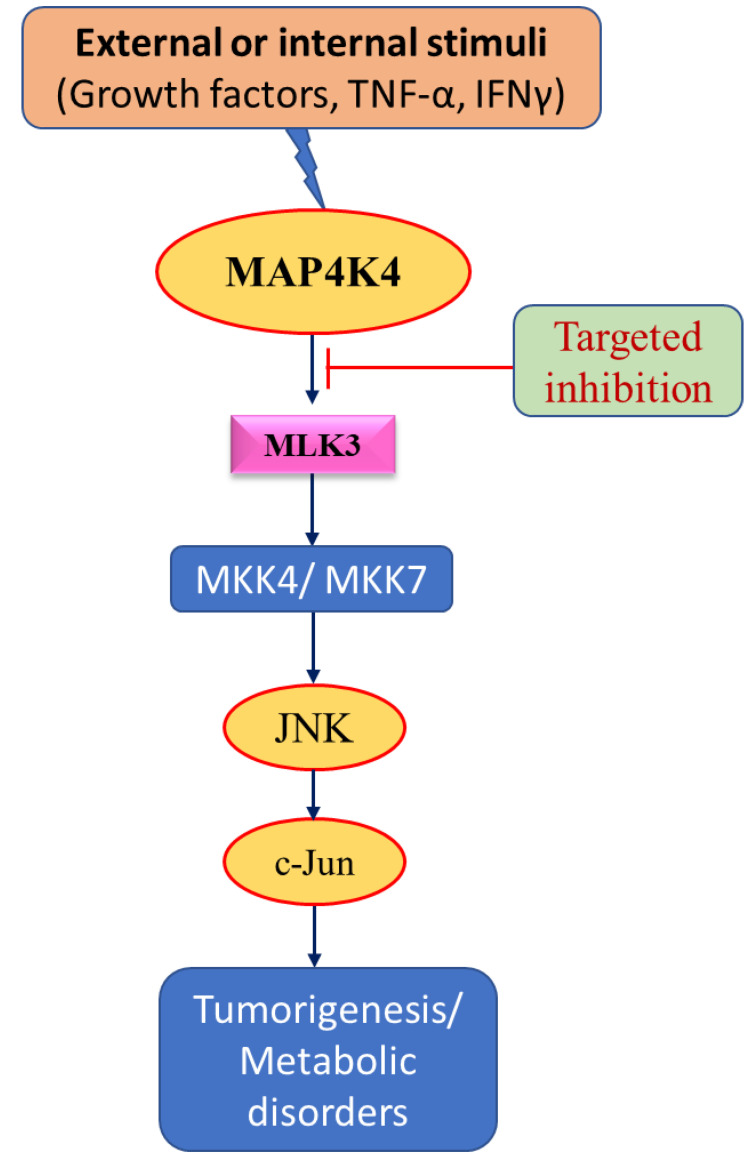
Schematic diagram showing therapeutic targets. Targeting MAPK signaling using inhibitors at specific sites to treat cancer and metabolic disorders and enhance patient survival.

**Table 1 cancers-15-02272-t001:** MAP4K4 involvement in metabolic disorders and different types of cancers.

Affected Cell/Tissue(s)	Inhibition Methods	Role and Effect	References
Colorectal cancer	RNAi or knockdown of MAP4K4 in vitro and in vivo	Regulation and progression of colorectal cancer	[71]
Hepatocellular carcinoma	Silencing of MAP4K4 with shRNA in HepG2 and Hep3B	↓ Cell proliferation, ↓ Blocked cell cycle progression at the S phase, ↑ Increased spontaneous apoptosis	[45]
Gastric cancer	Silencing of MAP4K4 by RNAi in BGC-823 cells	↑ Cell cycle arrest in the G1 phase, ↓ Cell proliferation	[76]
Lung adenocarcinoma	Induced mRNA transcript and protein levels of MAP4K4	MAP4K4 elevation was negatively associated with patients’ prognosis.	[78]
PDAC	Ectopic expression of miR-141 and knockdown of MAP4K4	↓ Tumorigenesis, ↓ Cell growth in vitro and in vivo, ↑ G1 arrest and apoptosis improved the chemosensitivity of pancreatic cancer cells.	[82]
PDAC	Inhibit MAP4K4 using GNE-495	↓ Tumor volume, ↓ Migration, ↑ Cell cycle arrest and ↑ survival	[23]
Endothelial cells	siRNA-mediated MAP4K4 depletion in endothelial cells	↓ Inflammatory capacity by reducing TNF-α-mediated permeability, ↓ TNF-α-induced leukocyte adhesion and leukocyte adhesion molecule expression	[61]
CD4^+^ T cell	MAP4K4 deletion using PMA and ionomycin	↓ CD4^+^ T cell proliferation	[67]

Abbreviations: MAP4K4: MAP kinase kinase kinase kinase; RNAi: RNA interference; shRNA: Short hairpin RNA; siRNA: Small interfering RNA; PDAC: Pancreatic ductal adenocarcinoma; PMA: Phorbol 12-myristate 13-acetate; TNF-α: Tumor necrosis factor-α; miR: MicroRNAs. ↓ indicates decrease, and ↑ indicates increase.

## Data Availability

The data can be shared up on request.

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
