# Peer review of "Molecular Insights of MAP4K4 Signaling in Inflammatory and Malignant Diseases"

_cancers, 2023, doi:10.3390/cancers15082272_

Round 1
Reviewer 1 Report (Previous Reviewer 1)
The Figures have been improved and updated in the revised manuscript. I have no more suggestions.
Reviewer 2 Report (Previous Reviewer 2)
Authors have incorporated all the suggestions.
This manuscript is a resubmission of an earlier submission. The following is a list of the peer review reports and author responses from that submission.
Round 1
Reviewer 1 Report
In the current review, Singh SK et al mainly reviewed the role of MAP4K4 in inflammatory and malignant diseases and its value in the targeted therapy. This review is well written and well reflects the significant function of MAP4K4 so far, based on a large number of relevant documents. However, it is better to optimize the Figures with professional graphics drawing software.
Reviewer 2 Report
In the present review article titled “Molecular Insights of MAP4K4 Signaling in Inflammatory and Malignant Diseases”, authors reviewed the potential role of
MAP4K4 in inflammatory and malignant diseases. No doubt the review is well written but most of the headings are already mentioned in the following reviews. https://pubmed.ncbi.nlm.nih.gov/27800153/
https://pubmed.ncbi.nlm.nih.gov/26791862/
https://pubmed.ncbi.nlm.nih.gov/27160798/
How this review is different from other published reviews?
There are only 250 paper showing on PubMed with keyword “MAP4K4”and in 2022, it is about 27 paper. Authors just cited only 3 recent papers (of year 2022). Authors should summarize all the paper related to MAP4K4. This will explore the wide role of MAP4K4 in human physiology and disease condition instead of presenting only on “Inflammatory and Malignant Diseases” (which is kind of already reviewed by other researchers).
